# Liposomal nanotheranostics for multimode targeted in vivo bioimaging and near-infrared light mediated cancer therapy

Rajendra Prasad [1], Nishant K. Jain [1], Amit S. Yadav[2,3], Deepak S. Chauhan[1], Janhavi Devrukhkar[1], Mukesh K. Kumawat[1], Shweta Shinde[1], Mahadeo Gorain[2], Avnesh S. Thakor[4], Gopal C. Kundu[2,3], João Conde [5✉] & Rohit Srivastava [1✉]

Developing a nanotheranostic agent with better image resolution and high accumulation into solid tumor microenvironment is a challenging task. Herein, we established a light mediated phototriggered strategy for enhanced tumor accumulation of nanohybrids. A multifunctional liposome based nanotheranostics loaded with gold nanoparticles (AuNPs) and emissive graphene quantum dots (GQDs) were engineered named as NFGL. Further, doxorubicin hydrochloride was encapsulated in NFGL to exhibit phototriggered chemotherapy and functionalized with folic acid targeting ligands. Encapsulated agents showed imaging bimodality for in vivo tumor diagnosis due to their high contrast and emissive nature. Targeted NFGL nanohybrids demonstrated near infrared light (NIR, 750 nm) mediated tumor reduction because of generated heat and Reactive Oxygen Species (ROS). Moreover, NFGL nanohybrids exhibited remarkable ROS scavenging ability as compared to GQDs loaded liposomes validated by antitumor study. Hence, this approach and engineered system could open new direction for targeted imaging and cancer therapy.

[1] Department of Biosciences and Bioengineering, Indian Institute of Technology Bombay, Powai, Mumbai, Maharashtra 400076, India. [2] Laboratory of Tumor Biology, Angiogenesis and Nanomedicine Research, National Center for Cell Science, Pune 411008, India. [3] School of Biotechnology and Kalinga Institute of Medical Sciences (KIMS), KIIT Deemed to be University, Institute of Eminence, Bhubaneswar 751024, India. [4] Interventional Regenerative Medicine and Imaging Laboratory, Department of Radiology, Stanford University, Palo Alto, CA 94304, USA. [5] Centre for Toxicogenomics and Human Health, Genetics, Oncology and Human Toxicology, NOVA Medical School, Faculdade de Ciências Médicas, Universidade Nova de Lisboa, 1169-056 Lisboa, Portugal. ✉email: joao.conde@nms.unl.pt; rsrivasta@iitb.ac.in

Nanotheranostics is an advanced development for localized imaging and cancer therapy[1–5]. So far, various conjugated nanohybrids have been proposed for noninvasive diagnosis and therapeutics, but their clinical translational is slow[6–16]. On the other hand, early stage tumor diagnosis and high accumulation of nanosized systems into a solid tumor environment are of crucial concern in onconanomedicine that needs to be addressed[14–20]. Traditionally used and previously attempted theranostics systems are facing several challenges, such as low image resolution, high toxicity, rapid clearance, poor and nonspecific biodistribution, low stability, fast aggregation, slow degradation, nonspecific biodistribution, shortage of multifunctional ability, and low tissue penetration ability[6,20–24]. Hence, these limitations hamper the preclinical investigations of nanobiomaterials in nanomedicine[14,20]. Thus, various integrated imaging, therapeutics, and targeting agents in a single platform, the so-called targeted theranostics agents, have been proposed for better outcomes[6,7,14]. However, integrating an "all in one platform" at nanoscale may reduce its diagnostic and therapeutic efficacy due to premature release of loaded diagnostic and therapeutic agents[24–29]. These pre-leaked cargo molecules may cause numerous side effects for healthy cells/tissues[30,31]. Further, the integration of all agents in one system can result in complicated synthesis routes with low product yield and low reproducibility[7,32–34].

In addition, the potential impact of enhanced permeability and retention effect, improving the uptake of nanohybrids into solid tumor microenvironment is a critical concern[7,14,20,35–37]. Second, targeting biomolecules improve the circulation and accumulation of injected nanoprobes for selective targeting to cancer cells, but are not yet convincing[6,38–40]. Now, the question about how to improve the accumulation of nanosized particles into solid tumor environment is the main focus of the present work. Further, it has been noticed that the external field or stimuli, such as near-infrared (NIR) light, temperature, magnetic field, etc., may improve the uptake of nanoparticles; however, these are tested only up to cellular level, and very few reports are available at the in vivo level[7,41,42]. Presently, computerized tomography (CT) and emissive contrast imaging have been realized as the most convincing imaging approaches among all imaging modalities due to high electron and radiodensity of the used contrast agent (only for heavy metal contrast agents) in CT and the strong fluorescence ability of emissive probes (especially for photostable emissive agents) that exhibits ineffective tissue penetration in NIR range[6,7,21,43].

In terms of therapeutic strategy, several treatment plans have been applied for cancer cell ablation. Among them, NIR light-responsive phototriggered strategy (known as photothermal therapy, PTT) has emerged as a dynamic therapy for localized treatments using a variety of nanotheranostic agents[6,7,38,44]. In PTT, photothermal agents produce heat through surface plasmon resonance and electron–hole delocalization under NIR light irradiation[45–48]. Recently, apart from heat, the generation of reactive _cioxygen species (ROS) has been noticed as a side product of PTT[7,42,49]. Hence, the combined effect of generated heat and ROS from a single system shows oxidative and thermal damage of treated cancer cells/or tumor[7,42]. However, the produced nonspecific and uncontrolled ROS, and heat affect the surrounding healthy tissues that may cause inflammation, eschars, mutation, protein denaturation, cell apoptosis or necrosis, and mitochondrial dysfunctions[42]. To avoid the above limitations, and to improve the potential effect of PTT, the ROS scavengers[50] must be delivered precisely in the heterogeneous tumor environment that is specific for the target site/or ROS-enriched area. With this concept, few nanosized scavenger systems, especially platinum-coated gold nanorods (Pt-GNRs), have

been tried due to their good catalytic oxygen reduction ability during NIR light exposure[50]. Further, these Pt-GNRs prevent the oxidative damage of healthy cells under the NIR light irradiation. Apart from ROS scavenger ability, these Pt-GNRs have been studied for PTT at the in vitro level[50]. However, the ROS formation during PTT and its scavenging mechanism by plasmonic nanohybrids are not yet clear that is under investigation[42,50].

Herein, we report a liposomal nanotheranostics loaded with gold nanoparticles (AuNPs) and emissive graphene quantum dots (GQDs) named as NFGL. Further, an anticancer drug is loaded, and the NFGL surface is functionalized with folic acid (FA)-targeting ligands. Engineered nanohybrids demonstrate imaging bimodality for tumor diagnosis and site-selective tumor reduction during single-wavelength NIR light (750 nm) irradiation due to the combination effect of chemotherapy-photothermal therapy (chemo-PTT). Remarkably, ROS has been observed during NIR light exposure that is scavenged by targeted NFGL nanohybrids ensuring the catalytic effect of loaded plasmonic AuNPs.

## Results

### Stepwise assembly of multimode liposomal nanotheranostics.
Two different imaging probes, viz., AuNPs and GQDs, were integrated with liposomes (named as NFGL) using solvent-evaporated thin-film hydration method. The surface of liposomal nanohybrids was decorated with FA-targeting ligand that was further investigated for site-selective tumor imaging (contrast and fluorescent imaging modality) and NIR light-mediated tumor reduction (see Fig. 1a, b). First, the engineered liposomal nanohybrids were analyzed by low-beam voltage (100 kV) transmission electron microscopic (TEM) measurements exhibiting about 50 nm in size (see Fig. 2). Very few particles were observed with particle sizes between 60 and 100 nm. Further, better dispersion and particle-size distribution of prepared NFGL, FA attached NFGL, and drug-loaded FA attached NFGL nanohybrids were measured with dynamic light scattering (DLS), microscopic, and aqueous dispersion measurements shown in Supplementary Fig. 1. Prior to a detailed description of NFGL nanohybrids, herein, highly aqueous dispersible fluorescent and NIR-active GQDs were obtained from ethanolic extracts of *Mangifera indica* (mango) leaves through microwave-assisted synthesis route[51]. Obtained GQDs were with better size distribution and clear fringes as shown in Supplementary Fig. 2a, b. GQDs showed thickness of about 0.9–1 nm as measured through AFM (see Supplementary Fig. 2c). Next, the polymer-stabilized AuNPs as ROS scavengers were synthesized by using an earlier reported procedure with some modifications[52]. Controlled size distribution of synthesized AuNPs was seen through TEM imaging measurement as shown in Supplementary Fig. 2d. The encapsulation of tiny particles (GQDs, ~25% and AuNPs, ~33%) in the liposomal cavity were clearly observed from TEM images (see Fig. 2a–c). These loading capacities were analyzed using optical density measurements (see Supplementary Eq. 1). The selected-area electron diffraction (SAED) pattern of NFGL indicated the successful entrapment of the AuNPs (see Fig. 2d). Parent liposomal nanostructures were shown in Fig. 2e. Interestingly, the drastic contrast difference between empty and cavity-filled liposomes was clearly observed from TEM images that indicated the success of nanohybrids preparation.

Moreover, the elemental components, such as nitrogen (N), phosphorous (P), oxygen (O), and gold (Au) of NFGL, were analyzed through elemental analysis (elemental mapping and energy dispersive X-ray analysis, EDAX), confirming the encapsulation of AuNPs as shown in Fig. 3 and Supplementary Fig. 3. Further, surfactant directed process that described the

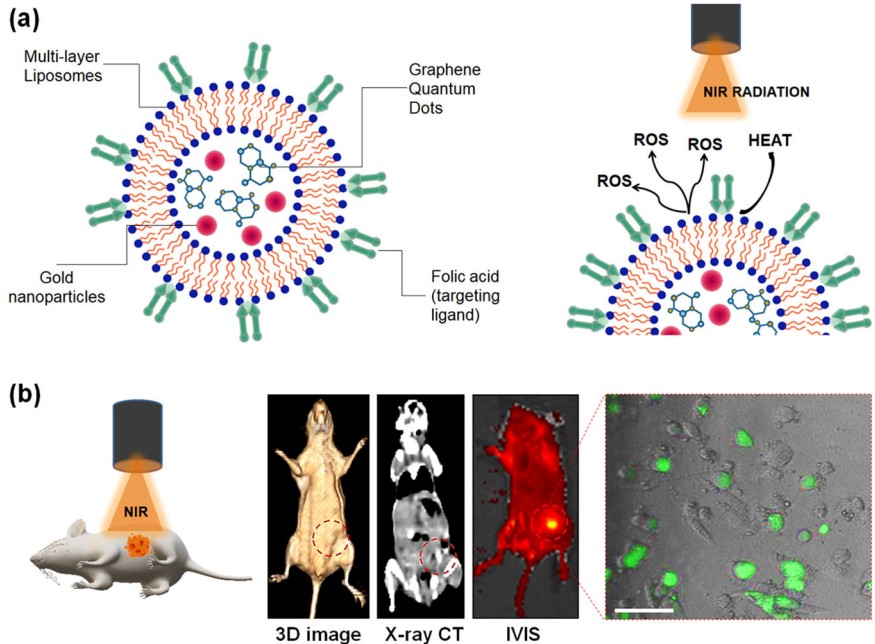

**Fig. 1 Schematic representation of liposomal nanotheranostics for multimode-targeted bioimaging and phototriggered cancer therapy. a** Folic acid targeting ligand decorated self-assembled liposomal nanohybrid loaded multimode imaging probes, viz., gold nanoparticles (AuNPs as radiocontrast for X-ray computed tomography and reactive oxygen species scavenger) and graphene quantum dots (GQDs as fluorescent contrast for near-infrared fluorescence imaging and photothermal agent). Designed functional liposomal nanohybrids demonstrating photothermal response/heat and the generation of reactive oxygen species (ROS, considered as the side product of photothermal therapy) under near-infrared (NIR) light exposure. **b** NIR light mediated cancer therapeutic representation with tumor-bearing mice model using engineered liposomal nanotheranostic agents and targeted imaging bimodality of breast cancer through X-ray computed tomography (X-ray CT) and in vivo imaging system (IVIS). Liposomal nanotheranostics treated cancer cells displaying the production of ROS (green emission represents the presence of ROS captured by DCFDA (2′,7′-dichlorofluorescin diacetate) dye staining) during NIR light exposure, scale bar = 20 µm.

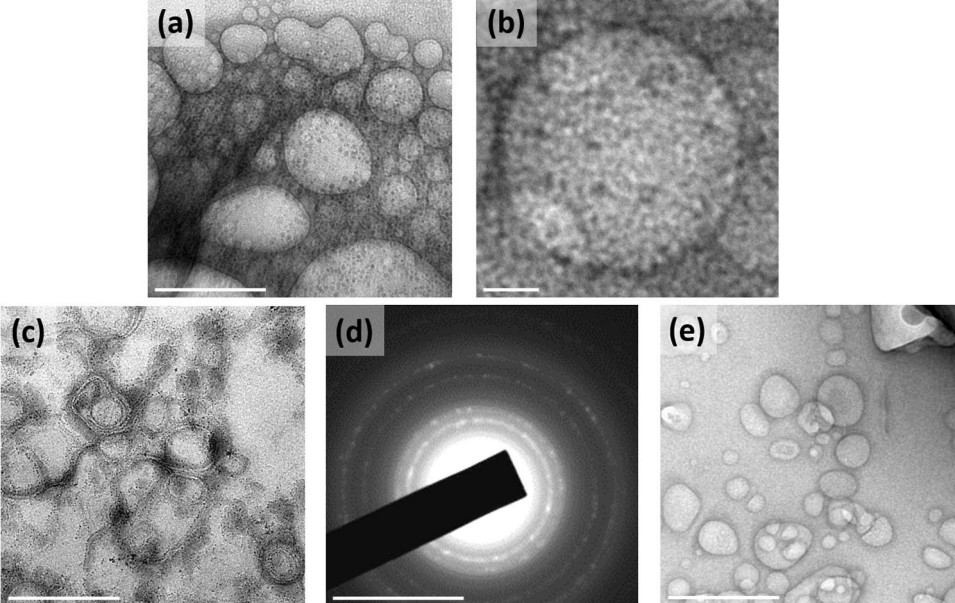

**Fig. 2 Transmission electron microscopic (TEM) images of engineered liposomal nanotheranostics showing spherical morphology. a**, **b** TEM images showing the successful encapsulation of graphene quantum dots (GQDs) in liposomal cavity, scale bar = 100 nm and 10 nm. **c** TEM imaging observation of gold nanoparticles (AuNPs) and graphene quantum dots (GQDs) encapsulated with liposomal nanohybrids named as NFGL, scale bar = 500 nm. **d** Selected area electron diffraction (SAED) pattern of NFGL nanohybrids, scale bar = 100 nm. **e** Microscopic image showing the distribution of parent liposomes (loaded with AuNPs and GQDs) with maintained spherical morphology, scale bar = 200 nm.

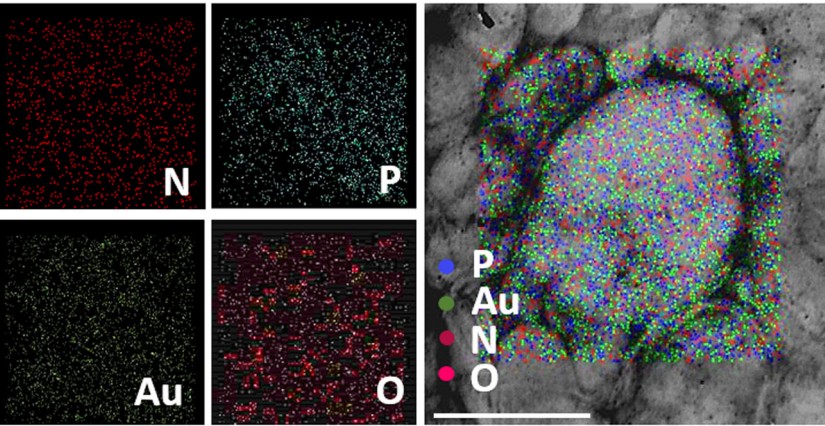

**Fig. 3 Microscopic elemental mapping of engineered liposomal nanotheranostics.** Elemental composition of gold nanoparticles (AuNPs) and graphene quantum dots (GQDs) encapsulated with liposomal nanohybrids (NFGL) analyzed through transmission electron microscopic (TEM) images showing the presence of nitrogen (N in maroon color), phosphorous (P in blue color), gold (Au in emerald color), and oxygen (O in pink color) elements with individual and merged imaging, scale bar = 300 nm.

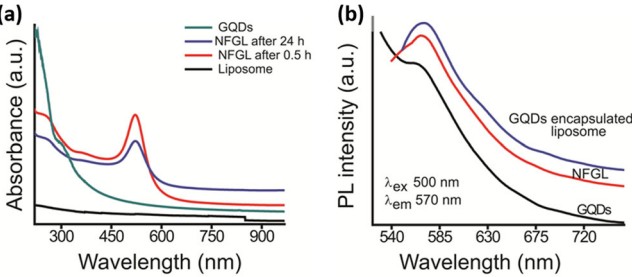

**Fig. 4 Optical properties of designed NFGL nanohybrids validated through spectroscopic measurements. a** Absorption spectra of parent liposome (loaded with AuNPs and GQDs), prepared graphene quantum dots (GQDs), and gold nanoparticles (AuNPs) and graphene quantum dots (GQDs) loaded liposomal nanohybrids named as NFGL at two different time points, viz., 0.5 h and 24 h, indicating the presence of multimode probes (GQDs and AuNPs) within liposomal particles. **b** Photoluminescence spectra of prepared graphene quantum dots, GQDs encapsulated liposomes, and engineered NFGL nanohybrids using 500 nm excitation wavelength, demonstrating better emissive property of fabricated nanotheranostics.

mixing of silver nitrate ($AgNO_3$) with $HAuCl_4$ (gold precursor) in the presence of ascorbic acid (reducing agent) produced monodispersed AuNPs that were characterized by TEM and EDAX. Negative ζ potentials of −44 mV and −38 mV were obtained for GQDs and AuNPs due to the presence of –OH and –SH functional groups of quantum dots and AuNPs. Liposomes showed the positive ζ potential of 28 mV due to a cationic choline head group of lipid molecules, whereas NFGL showed ζ potential of 10 mV, ensuring the interaction of GQDs and AuNPs with lipid self-assembly (see Supplementary Fig. 4). From RAMAN spectra, D and G bands of carbon framework confirmed the presence of GQDs (1328 and 1590 $cm^{-1}$) in the NFGL (1323 and 1587 $cm^{-1}$) as shown in Supplementary Fig. 5.

**Optical properties of liposomal nanotheranostics.** The optical properties of NFGL were analyzed through absorption and photoluminescence spectroscopic measurements (see Fig. 4). The absorption spectra of NFGL showed a sharp band at ~524 nm, indicating the transverse peak of loaded AuNPs that was further stable even after 24 h, demonstrating the better photostability of encapsulated AuNPs (see Fig. 4a). The peak sharpness revealed the high dispersion of AuNPs. Moreover, loaded AuNPs in NFGL

were confirmed through TEM images and elemental analysis (as shown in Figs. 2c and 3). Contrasting ability of NFGL nanohybrids was evaluated and discussed in the present work due to high electron density and atomic number of these encapsulated AuNPs (discussed below). In addition, the hydrophilic cavity of liposomes was loaded with emissive GQDs that demonstrated broad absorption in the NIR region (700–900 nm), indicating their applicability for phototherapy. Peaks of unsaturated carbonic frameworks in GQDs (C=C and C=O) were noticed between 200 and 300 nm (see Fig. 4a). Emission spectra of NFGL, GQDs loaded parent liposomes, and purified GQDs, indicated the orange-yellow emissive nature of fabricated nanohybrids (570 nm emission at 500 nm of excitation) due to the surface defects and functional groups of encapsulated GQDs as shown in Fig. 4b. In addition, the liposomal cavity of nanohybrids was also loaded with anticancer drug doxorubicin hydrochloride that was confirmed through spectroscopic analysis where the absorption of doxorubicin hydrochloride was observed at around 493 nm (shown in Supplementary Fig. 6).

**Multimodal characteristics of liposomal nanotheranostics.** Entrapped AuNPs induced the contrast modality of the fabricated NFGL nanohybrids due to high electron density and high atomic number of Au. Radiodensity of NFGL was measured at various concentrations (5–100 μg/mL) using clinical X-ray computed tomography (X-ray CT, TOSHIBA 64 CT scanner, 120 kVp tube voltage, 250 mA tube current, 5 mm slice thickness, and 1 s rotation time) scanner showing a linear correlation between contrast and concentration of nanohybrids (see Fig. 5a). Radiodensity and brightness were analyzed by recording the Hounsfield units (HU) for the selected area of interest followed by RadiAnt DICOM Viewer software. The second imaging modality, viz., NIR fluorescence (NIRF) of engineered NFGL nanohybrids, was measured by using the in vivo imaging system (IVIS) at 500 nm of excitation wavelength. The aqueous suspension (100 μg/mL) of NFGL demonstrated the promising emission property due to the photoluminescence ability of loaded photostable GQDs as shown in Fig. 5b (emission from sample contained tubes shown in the inset). Phosphate-buffered saline (PBS) and parent liposomes were used as control during imaging measurements.

Importantly, the unsaturated carbonic framework in GQDs made them photothermally active under NIR light irradiation. About 750 nm wavelength of the NIR laser (1 W/$cm^2$) was applied to conduct the phototransduction of GQDs, and NFGL

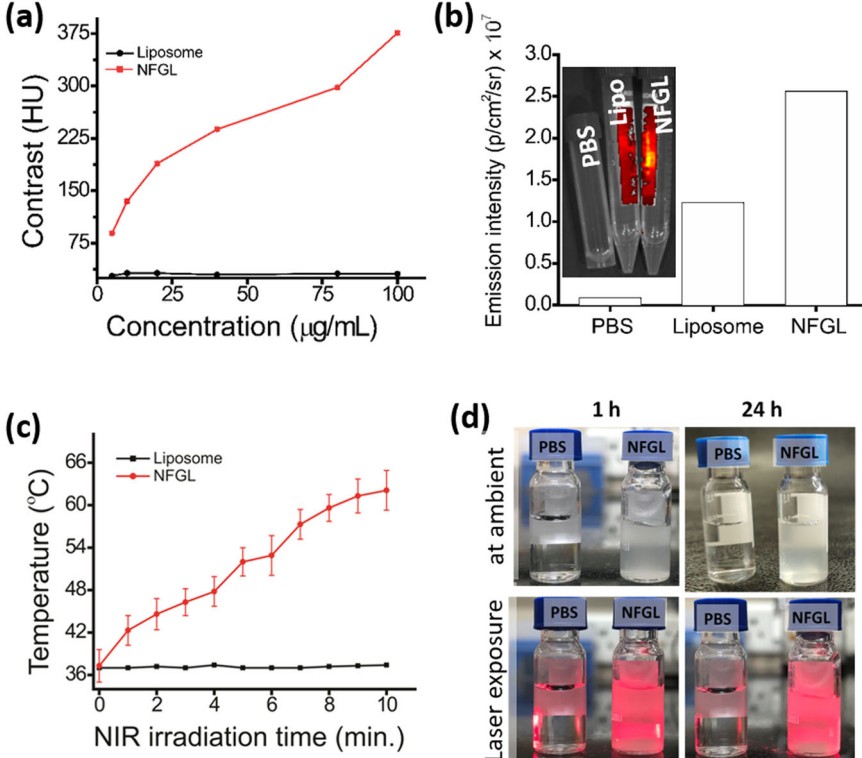

**Fig. 5 Multimodal characteristics of liposomal nanotheranostics. a** Contrast measurements (also known as radiodensity) of designed gold nanoparticles (AuNPs) and graphene quantum dots (GQDs)-loaded liposomal nanohybrids named as NFGL at various concentrations (5–100 µg/mL) using a clinical TOSHIBA 64 CT clinical scanner with 5 mm slice thickness and 1 s rotation time compared with parent liposome (loaded with AuNPs and GQDs), revealing the presence of AuNPs (high electron coefficient and density) within the liposomal framework. **b** Emission performance of NFGL and compared with parent liposomes and PBS using the in vivo imaging system (IVIS) showing the better contrast ability for deep tissue penetration, and indicating the presence of GQDs within liposomal assembly. **c** Time-dependent photothermal transduction performance of NFGL nanohybrids at 0.5 mg/mL concentration using 750 nm of NIR light irradiation (1 W/cm²) compared with parent liposome (n = 3), ensuring the potential impact of phototriggered therapy. **d** Digital photographs showing dispersion of NFGL at ambient conditions, and during laser exposure after 1 h and 24 h of time periods.

nanohybrids showed a regular rise in temperature with respect to NIR exposure time. Hyperthermia temperature (43 °C) was noticed in 3 min, whereas ablation temperature (~49.9 °C) was recorded within 10 min of light irradiation using 0.5 mg/mL concentration of NFGL that was compared with parent liposomes, indicating the better NIR activity of encapsulated GQDs (see Fig. 5c). Moreover, at 0.2 mg/mL concentration of NFGL, the temperature reached ~43.3 °C after 3 min of NIR exposure that was further raised up to 55 °C after 10 min of light exposure. About 1 mg/mL concentration of NFGL showed maximum temperature of about 60 °C after 10 min of NIR light exposure (see Supplementary Fig. 7). Similarly, GQDs loaded liposomes showed good photothermal response that was due to the loaded GQDs. In light of heat-response measurements, the temperature reached 43.2 °C in 4 min at 0.2 mg/mL concentration, and that was further recorded to about 54 °C in 10 min of exposure. Quick hyperthermia temperature (~43.3 °C in 2 min) was recorded at 0.5 mg/mL concentration of nanohybrids, and the maximum temperature of about 59 °C in 10 min was recorded at 1 mg/mL concentration of nanohybrids. The temperature was stabilized at 37 °C for negative control (PBS and liposomes) as shown in Supplementary Fig. 7. Next, the phototriggered drug release performance of DOX-loaded NFGL was examined in various environments. Before NIR exposure, negligible drug release (about 3.5%) was obtained in physiological conditions (pH 7.4), whereas more than 70% was observed after NIR irradiation during 24 h of incubation time due to produced heat by loaded GQDs. In case of acidic environment (cancer cell interior

environment, pH 3.0), about 77% of drug released was noticed within 6 h even without NIR light exposure that was further achieved ~98% after NIR light exposure (only one shot of 10-min NIR exposure), indicating the combined effect of chemo-PTT (see Supplementary Fig. 8). In the disintegration test, controlled morphology of nanohybrids was seen in the physiological condition before NIR irradiation, whereas disintegrated particles were noticed after NIR exposure due to generated heat from encapsulated GQDs as shown in Supplementary Fig. 9. Further, in acidic environment (pH 3.0), the degradation of the designed system was evidently noticed due to protonation of lipid and GQDs that revealed the destabilization of lipid self-assembly. Interestingly, the complete disintegration of nanohybrids was observed in acidic environment (pH 3.0) under the NIR light exposure. The colloidal stability of designed nanohybrids was ensured by dispersion examinations before and after laser exposure (750 nm) at 1 h and 24 h of incubation time, showing high dispersion without any turbidity (see Fig. 5d). In addition, the aqueous dispersion of designed NFGL, DOX-loaded NFGL, and GQDs loaded liposomal nanohybrids was tested up to 1 week of time period as shown in Supplementary Fig. 1. Hence, these examinations confirmed the multifunctionality of designed liposomal nanohybrids.

**In vitro evaluation of liposomal nanotheranostics**. To evaluate the intracellular localization and targeted cancer therapeutics, FA targeting ligand was attached to the NFGL nanohybrids using polyethylene glycol (NH₂–PEG) linker[53]. FTIR spectra revealed

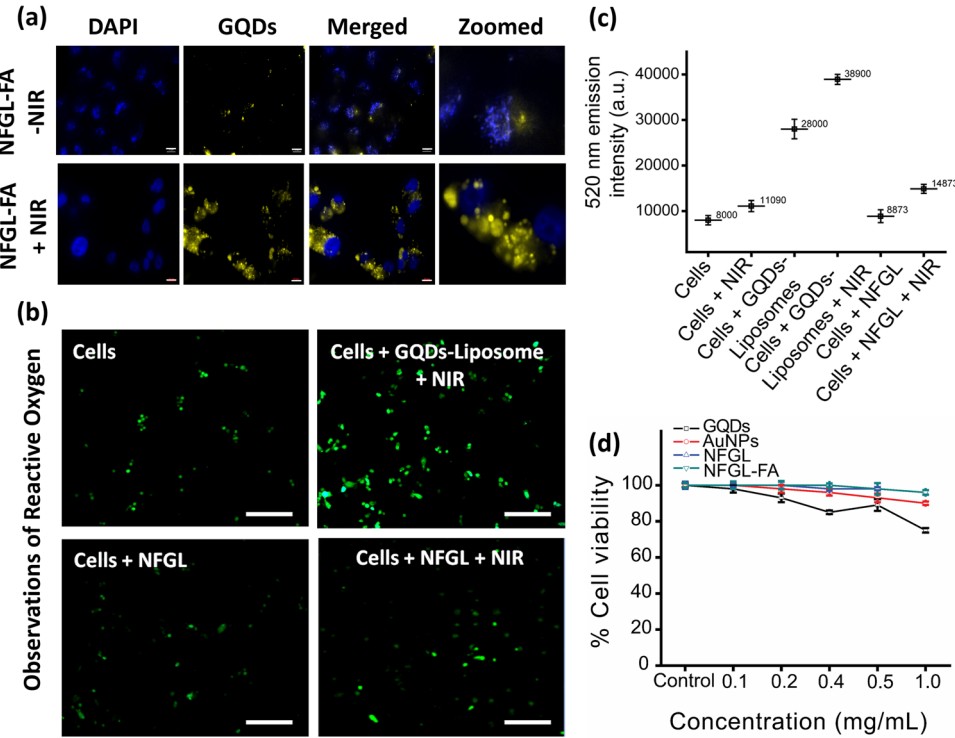

**Fig. 6 In vitro validations of formulated gold nanoparticles (AuNPs) and graphene quantum dots (GQDs) loaded liposomal nanotheranostics. a** Cancer cell imaging and cellular uptake of folic acid functionalized NFGL nanotheranostic agents (NFGL–FA) with 4T1 breast cancer cell lines with and without NIR light exposure (750 nm, 1 W/cm$^2$ for 10 min of exposure), scale bar = 10 μm. **b** Observations of produced reactive oxygen species (ROS, as a side product of photothermal therapy) during NIR light irradiation using various formulations of NFGL nanohybrids treated with 4T1 cancer cell lines; green emissive ROS are noticed by (2′,7′-dichlorofluorescin diacetate, DCFDA) dye staining. **c** Quantitative analysis of produced ROS from nanohybrids treated with breast cancer cells with and without NIR light irradiation (n = 3). **d** Percentage cell viability of NFGL nanohybrids before and after folic acid functionalization and its major components (GQDs and AuNPs) using 24 h MTT assay at different concentrations (0.1–1 mg/mL, n = 3).

that the O–H stretching vibrations at 3376 cm$^{-1}$, C=O stretching vibrations, and N–H bending at 1650 and 1587 cm$^{-1}$ were assigned to the CONH group. Peaks between 1640 and 1655 cm$^{-1}$ were attributed to the bending of NH$_2$, and stretching vibrations of CH$_2$ obtained at 2947 cm$^{-1}$ confirmed the presence of NH$_2$–PEG. Further, the bands between 1390 and 1550 cm$^{-1}$ indicated aromatic ring stretch of pteridine and $p$-amino benzoic acid moieties of FA[6,7]. Thus, FTIR spectra confirmed the presence of oxygen rich functional groups on GQDs and surface functionalization of NFGL nanohybrids (see Supplementary Fig. 10a, b). Next, 4T1 breast cancer cell lines were treated with emissive nanohybrids before and after NIR exposure. Enhanced distribution of nanohybrids in the cancer cell interior was observed after NIR irradiation as compared with without NIR treatment, may indicate the damage of cellular membrane by generated heat (6 h of fluorescence imaging, Fig. 6a). On the other hand, promising cellular uptake (6 h of incubation time) of NFGL nanohybrids with 4T1 cells was achieved with the help of FA functionalization (NFGL–FA) as shown in Supplementary Fig. 11.

Recently, ROS has been noticed as the side product of light-mediated photothermal strategy that damages the healthy cells via oxidizing the cellular matrix[7,42]. Similarly, we observed the presence of ROS along with photothermal heat during NIR light exposure (750 nm) due to the hydroxyl and carboxyl groups of loaded GQDs in liposomal nanohybrids (see Supplementary Fig. 12). Remarkably, the NFGL nanohybrids showed minimal appurtenance of ROS from cancer cell interior during NIR light exposure, maybe due to the ROS scavenging effect[50] of loaded AuNPs. Further, the presence of ROS was detected by green

(2′,7′-dichlorofluorescin diacetate, DCFDA) dye staining that was confirmed through fluorescence microscopic images as shown in Fig. 6b, c. The homogeneous and deep intracellular distribution of ROS from cancer cell interior was clearly noticed during NIR light exposure that allowed the easy uptake of nanoparticles as shown in Supplementary Fig. 13. However, the clear mechanism of ROS generation during NIR light irradiation experiments is unknown and under investigation.

Next, 24 h MTT measurements in NIH-3T3 cell lines showed more than 90% cell viability for GQDs, AuNPs, NFGL, and FA-functionalized NFGL (0.1–1 mg/mL, Fig. 6d). Further, in vitro therapeutics examinations were carried out on 4T1 and MCF-7 cancer cell lines using different formulations of nanohybrids as shown in Supplementary Figs. 14 and 15, respectively. In 4T1 cells, FA-attached GQDs (GQD-FA) and NFGL (NFGL–FA) showed noteworthy cancer cell death (about 80%) under NIR light treatment due to the photothermal and oxidative damage that was about 89% in the case of DOX-loaded NFGL–FA under NIR light exposure due to the combined chemo-PTT effect. Before NIR light exposure, these systems showed negligible cell death (18–25%). Whereas, non-targeted NFGL demonstrated minimum cell death (32%) due to poor cellular uptake. Interestingly, only NIR light and NFGL nanohybrids showed ~98% cell viability that was comparable with untreated cancer cells (about 100% considered as control group). In MCF-7 cells, nanohybrids showed more than 95% cell viability, whereas about 45 and 35% cell viabilities were observed after NIR light exposure due to the photothermal and ROS effect. DOX–NFGL showed ~70% cell viabilities before FA attachment, and dropped to 42%

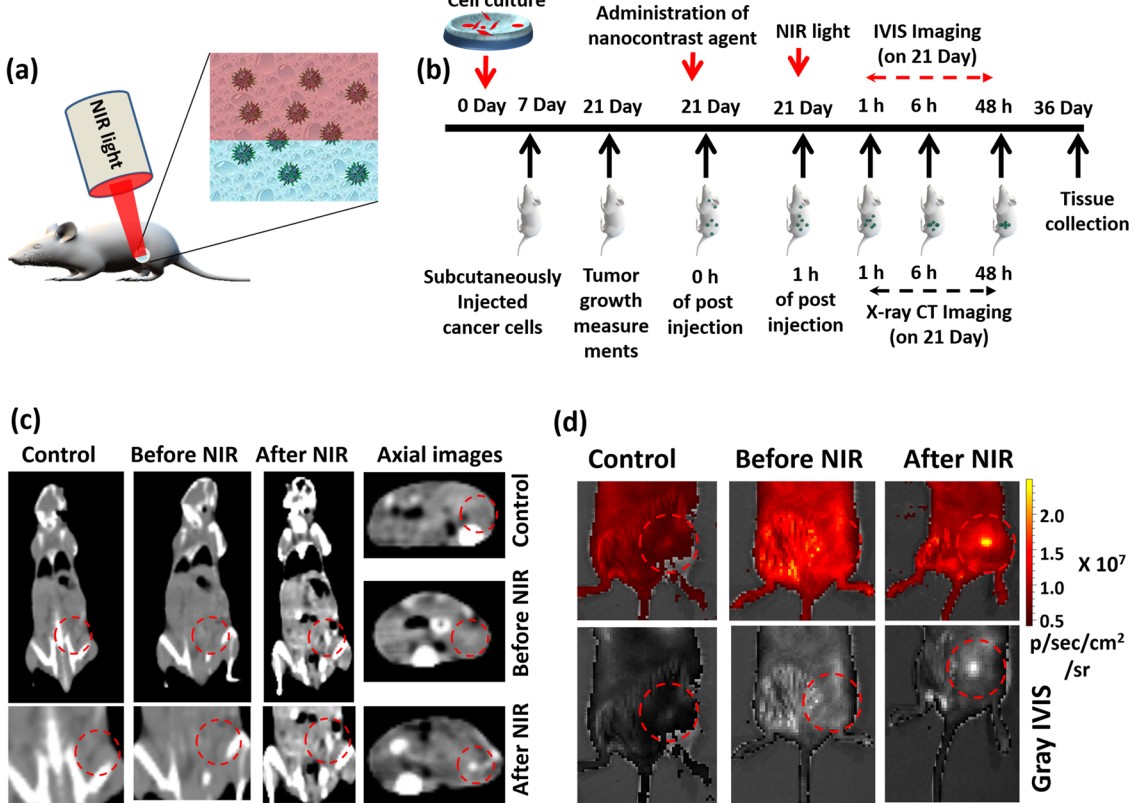

**Fig. 7 Site-selective multimode tumor imaging. a** Planned NIR light mediated phototriggered strategy for post-injected 4T1 tumor bearing mice showing enhanced tumor uptake of injected gold nanoparticles (AuNPs) and graphene quantum dots (GQDs) loaded liposomal nanotheranostics with folic acid functionalization (NFGL–FA). **b** Experimental flowchart from day 0 (cell culture) to tissue collection (36 days) followed by multimodal tumor diagnosis and biodistribution experiment setup. **c** Localized tumor diagnosis and specific biodistribution measurements after 48 h of time before and after NIR light exposure (750 nm, 1 W/cm$^2$ for 10 min) followed by whole body X-ray computed tomography scans with coronal and axial CT slices of mice body using TOSHIBA 64 CT scanner at 120 kVp tube voltage and 250 mA tube current with 5 mm slice thickness and 1 s rotation time. **d** Targeted deep tumor localization in mice body before and after NIR light exposure (750 nm, 1 W/cm$^2$ for 10 min) using the in vivo imaging system (IVIS). In both imaging modalities, pre-injected mice were considered as control groups.

after FA attachment, due to the targeted chemotherapeutic effect of NFGL nanohybrids. Moreover, only 10% cell viability was observed for targeted combined chemo-PTT (DOX–NFGL–FA under NIR light exposure). Hence, we believed that the NIR light irradiations improve the efficacy of heat and ROS that enhance the uptake of nanoparticles through photothermal and oxidative rupture of cancer cell membrane.

**Imaging bimodality for tumor diagnosis and biodistribution.** In the present work, we established a NIR light triggered strategy for enhanced accumulation of injected NFGL based nanotheranostic agents into 4T1 breast tumor bearing mice models as shown in Fig. 7a. FA attached NFGL (NFGL–FA) was tested as a safe multimode contrast agent (X-ray CT and NIRF imaging) for localized tumor diagnosis (see Fig. 7b). After 2 weeks of tumor growth, a single dose of nanohybrids was subcutaneously administered (100 µl) at the tumor site. After 1 h post injection of NFGL–FA, the tumor area was exposed with 10 min of NIR light (750 nm, 1 W/cm$^2$), and examined for X-ray CT and NIRF imaging as shown in Fig. 7c, d, respectively. In vivo X-ray CT images of NIR treated animals exhibited higher brightness and contrast from tumor area that were much better as compared with without NIR treated and pre-injected mice. The enhanced contrast from tumor area was maintained even after 48 h of post injection, indicating the strong binding and higher accumulation ability of injected liposome based nanocontrast agents (see

Fig. 7c). These obtained outcomes were corroborated with NIRF imaging modality that revealed the improved emission intensity from NIR light exposed tumor area as compared with without NIR treated and pre-injected mice as shown in Fig. 7d.

Further, the designed targeted nanohybrids were tested for biodistribution and tumor accumulation measurements at various time points (1, 6, and 48 h) followed by subcutaneous post injection as shown in Fig. 8 and Supplementary Fig. 16. Whole body scans of NFGL–FA-injected animals were captured by using imaging bimodalities (CT and IVIS) that were compared with the control group (untreated/pre-injected animals). In biodistribution measurements of NIR treated animals, the coronal and axial CT slices of mice body showed the higher radiodensity and contrast (HU values) from tumor (~243 HU), heart (~140 HU), liver (about 210 HU), spleen (68 HU), intestine (~79 HU), and kidneys (~42 HU) that were considerably different from the radiodensity of NIR exposed animal's organs like heart (~110 HU), liver (about 198 HU), spleen (50 HU), intestine (~52 HU), kidneys (~39 HU), and tumor (~200 HU) and pre-injected mice as shown in Fig. 8a–c. Importantly, the contrast and brightness decreased gradually from the upper abdominal organs and accumulated in the lower abdomen organs with higher contrast and brightness with respect to post-injection time (6 h and 48 h) in all cases as shown in Supplementary Table 1. However, pre-injected animals demonstrated negligible contrast (in the range of 20–40 HU) from tumor area (38–40 HU for 6–48 h of time) and other major organs in all the cases. The drastic change in

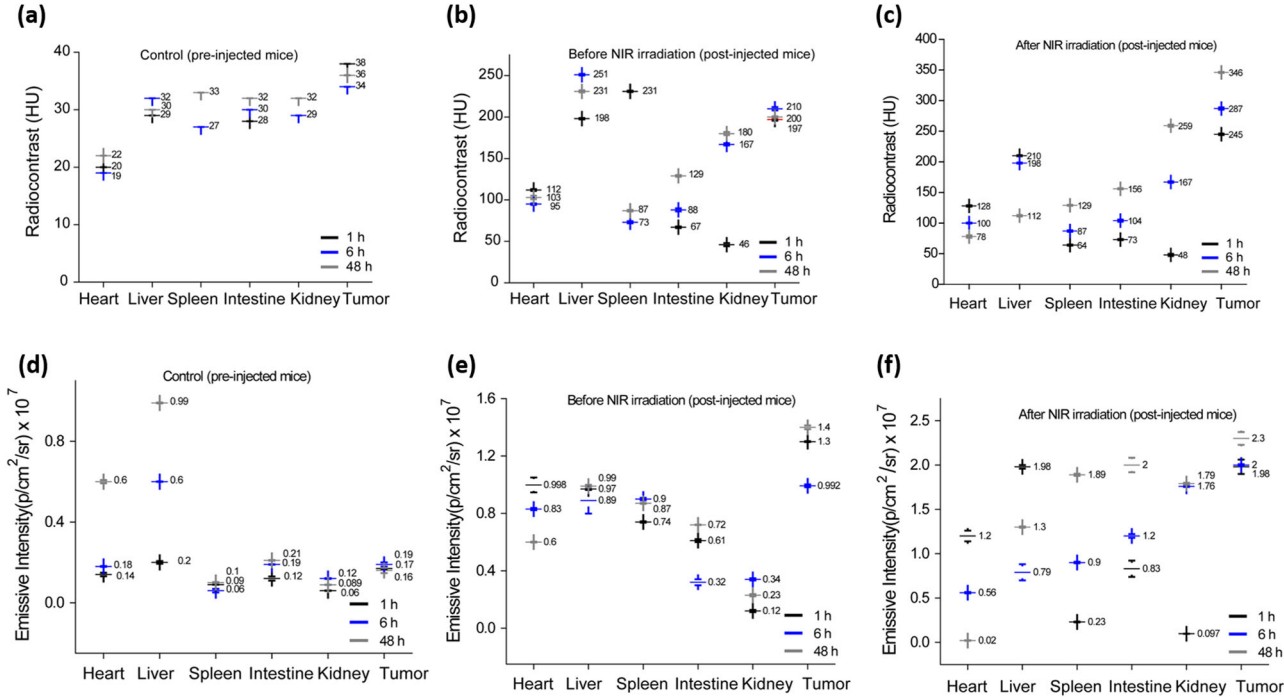

**Fig. 8 Multimode image analysis using in vivo X-ray computed tomography (X-ray CT) and near-infrared fluorescence imaging showing specific biodistribution with promising contrasting ability of post-injected gold nanoparticles (AuNPs) and graphene quantum dots (GQDs) loaded liposomal nanotheranostics with folic acid functionalization (NFGL–FA) in major organs and tumor. a–c** Quantitative analysis from X-ray CT imaging ($n = 3$ mice per group) and **d–f** near-infrared fluorescence imaging using the in vivo imaging system ($n = 3$ mice per group) with and without NIR exposure experiments and compared with pre-injected mice (considered as control groups).

radiodensity and brightness between pre-injected and without NIR exposure, and NIR treated post-injected tumor bearing mice, demonstrated the better accumulation and strong binding ability of NFGL–FA nanohybrids.

On the other hand, fluorescent based imaging measurements demonstrated the promising tumor accumulation and binding ability that were measured up to 48 h before and after NIR exposure (see Fig. 8d–f). After NIR light exposure, the momentous emission from the tumor site was noticed within an hour of NFGL–FA post injection that was observed up to 48 h, which was much better from pre-injected and without NIR treated animals. Biodistribution measurements of post-injected animals (1, 6, and 48 h) confirmed the smooth circulation and specific distribution of injected nanohybrids. Within 1 h of post injection, NIR treated mice demonstrated maximum emissive intensity from tumor ($1.98 \times 10^7$ p/s/cm²/sr), heart ($1.2 \times 10^7$ p/s/cm²/sr), liver ($1.98 \times 10^7$ p/s/cm²/sr), spleen ($0.28 \times 10^7$ p/s/cm²/sr), intestine ($0.64 \times 10^7$ p/s/cm²/sr), and kidneys ($0.18 \times 10^7$ p/s/cm²/sr) that was further increased in lower abdominal organs and on the tumor site ($2.0 \times 10^7$ p/s/cm²/sr and $2.3 \times 10^7$ p/s/cm²/sr) with respect to the course of time (6 h and 48 h) as given in Supplementary Table 2. We noticed a drastic difference in emission intensities between the pre-injected, without NIR exposure, and NIR treated animals that demonstrated the potential role of NIR light for enhanced tumor uptake of nanohybrids. Importantly, the contrast and emission enhancement in lower abdominal organs may reveal the clearance of small sized imaging agents.

To evaluate site-selective tumor imaging and specific biodistribution (1, 24, and 48 h), NFGL–FA was intravenously injected into tumor bearing mice (10 mg/kg body weight, single-dose administration of 100 µl). Maximum emission intensity from tumor area was achieved after 24 h of post injection, specifying the highest accumulation of nanohybrids in tumor

microenvironment that was further notable up to 48 h due to the strong binding ability of NFGL–FA (see Fig. 9a, Supplementary Fig. 17a). In biodistribution of injected nanohybrids, the fluorescent intensity from the heart, lungs, liver, kidneys, spleen, intestine, and tumor was measured after 48 h of post injection as shown in Fig. 9b. Moreover, remarkable emission intensity from tumor, intestine, and kidneys was noticed compared with the liver and lungs (see Supplementary Fig. 17b). Remarkable emission from tumor area corroborated the large uptake of nanohybrids in tumor microenvironment. Stimulatingly, there was no emission from the heart and spleen, and emission from the liver, intestine, and kidneys may indicate the easy excretion of injected nanohybrids.

**In vivo therapeutic evaluation for localized tumor reduction.** A comprehensive investigation of light to heat transduction, ROS scavenging performance, and in vitro anticancer efficacy ensured the suitability of prepared liposomal nanotheranostics for in vivo therapeutic studies. Next, five groups of tumor-bearing animals (three mice per group) were treated with various therapeutic conditions to evaluate the tumor ablation. In detail, a minimal dose (10 mg/kg body weight) of engineered nanohybrids, such as FA attached NFGL (NFGL–FA), doxorubicin hydrochloride loaded NFGL–FA (DOX–NFGL–FA), and GQDs loaded liposomes with FA attachment, was intravenously administered for phototriggered antitumor therapy and tumor ablation. In vivo therapeutic ability of designed nanohybrids was demonstrated on 4T1 tumor bearing mice for targeted chemotherapy (DOX–NFGL–FA), NIR light mediated site-specific phototherapy (NFGL–FA under NIR light exposure), and localized NIR light triggered combined chemophototherapy (DOX–NFGL–FA during 750 nm of NIR light irradiation) as shown in Fig. 9c–f. ROS scavenging performance at the in vivo level was tested by using NFGL–FA and FA attached GQDs loaded liposome

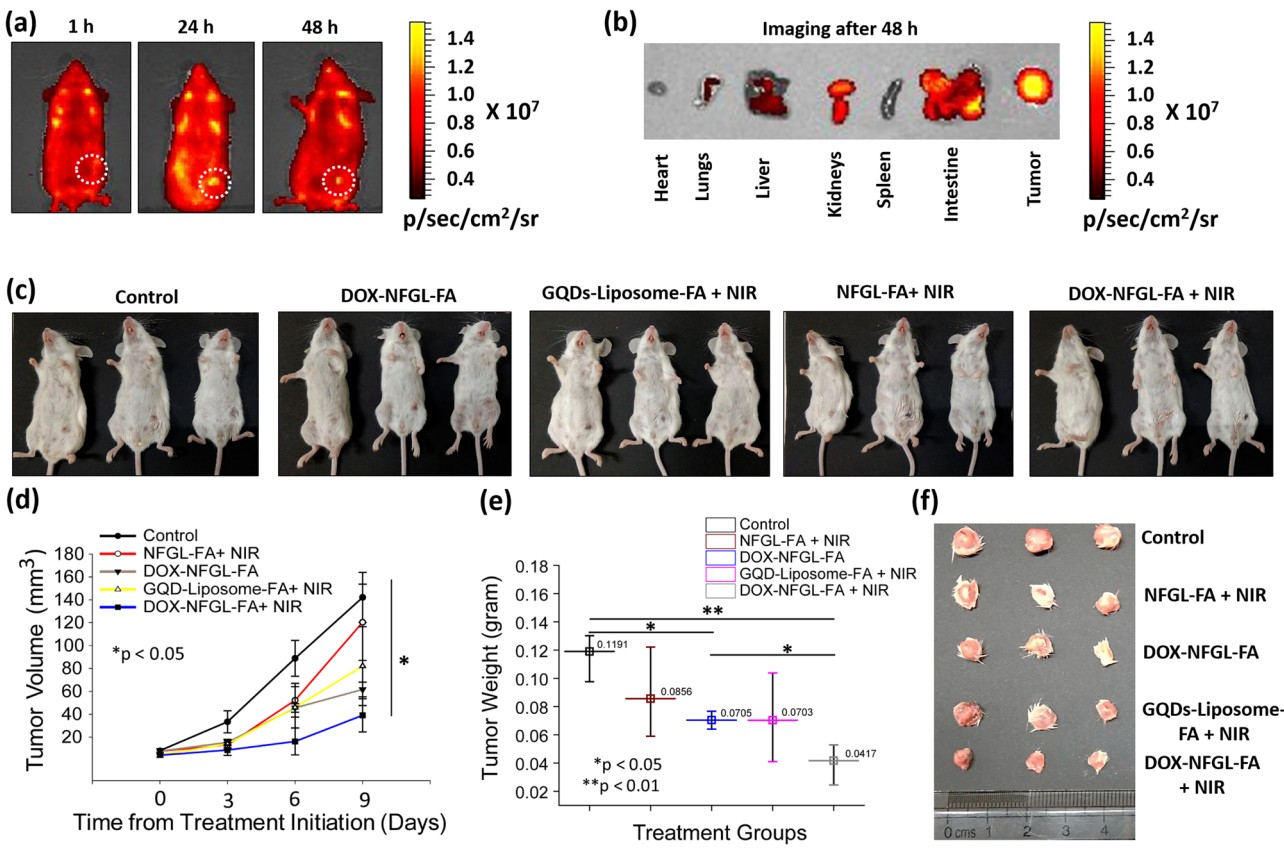

**Fig. 9 Localized tumor diagnosis and phototriggered tumor reduction measurements. a** Whole body in vivo imaging for site-selective 4T1 tumor diagnosis at various time points (1, 24, and 48 h) of intravenously injected gold nanoparticles (AuNPs) and graphene quantum dots (GQDs) loaded liposomal nanotheranostics with folic acid functionalization (NFGL–FA). **b** Ex vivo imaging of collected major organs and 4T1 tumor after 48 h from intravenously nanotheranostics injected animals. **c** Digital photographs of 4T1 tumor bearing mice during their therapeutic conditions after intravenous injection of NFGL–FA nanotheranostics ($n = 3$ mice per group) showing the successive tumor regression in various therapeutic conditions with good health of treated mice. **d, e** Measurements of tumor reduction by tumor volume ($mm^3$, $^*p < 0.05$) and tumor weight (gram, $^*p < 0.05$, $^{**}p < 0.01$) analysis ($n = 3$ mice per group) during various therapeutic conditions using different formulations of NFGL–FA nanotheranostics with and without NIR light exposure (750 nm, 1 W/cm² for 10 min), and compared with the control group of animals (pre-injected and untreated mice). **f** Digital photograph of collected tumors after various therapeutic assessments using different formulations of NFGL–FA nanotheranostics representing the promising tumor reduction and potential impact of phototriggered cancer therapy.

(GQDs–liposome–FA) nanohybrids in photothermal conditions. Especially, GQDs liposome–FA treated mice showed better tumor reduction compared with NFGL–FA treated mice during NIR light exposure (see Fig. 9c–e). The combined effect of produced ROS (a side product of PTT) and generated heat causes the tumor regression in the case of GQDs–liposome–FA, whereas the produced ROS was catalyzed by loaded AuNPs in the NFGL–FA nanohybrids during NIR light mediated antitumor activity resulting in low tumor reduction (see Fig. 9d, e). During in vivo therapeutic course, various qualitative and quantitative analyses of tumor volume, weight, animal health, and body weight were evaluated. A drastic reduction in tumor volume and weight was measured in the case of DOX–NFGL–FA (39.04 $mm^3$ and 0.041 g) during NIR light treatment due to the combined effect of the released anticancer drug and generated heat. Whereas, GQDs–liposome–FA (82.07 $mm^3$ and 0.085 g) treated mice showed promising tumor reduction due to the combined effect of produced ROS and heat (a key factor of PTT).

From in vivo ROS scavenging measurements during NIR light irradiation, the designed NFGL–FA nanohybrids showed better tumor shrinking (120.35 $mm^3$ and 0.070 g) compared with the control group (142.07 $mm^3$ and 0.119 g, pre-treated animals) due to the promising effect of generated heat as validated with the

obtained data given in Fig. 9d, e. In addition, there were no symptoms observed of eschars and inflammation on the animal's body during NIR light exposure, indicating the ROS scavenging ability of treated NFGL–FA nanohybrids (see Fig. 9c). Besides, DOX–NFGL–FA showed favorable tumor reduction (61.59 $mm^3$ and 0.070 g) without NIR light irradiation due to the impactful effect of targeted chemotherapy. Likewise, multiple therapeutic observations were described in the present work, demonstrating the reduction of 4T1 breast tumor (see Fig. 9f). Hence, multimode diagnostics and therapeutic observations indicated the potential impact of designed liposomal nanotheranostics for cancer treatments.

**In vivo toxicity of liposomal nanotheranostics.** In vivo toxicities were measured through histopathology analysis, hemolysis study, mice health, and body weight observations. The effect of injected nanohybrids on major organs like heart, liver, spleen, and kidneys, was evaluated through histopathology analysis at different time points (6 h and 48 h) as shown in Supplementary Fig. 18a. After 2 weeks, treated animals were sacrificed, and major organs were collected to examine the pathological changes in organs by hematoxylin and eosin (H&E) staining. H&E examinations demonstrated (1) healthy myofibers and muscle bundles in the

**(a)**

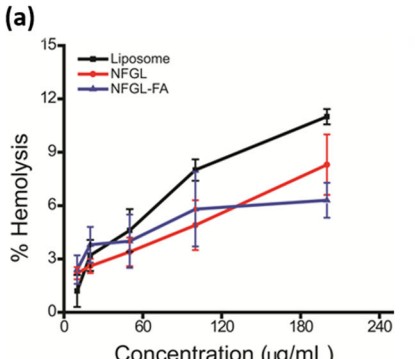

**(b)**

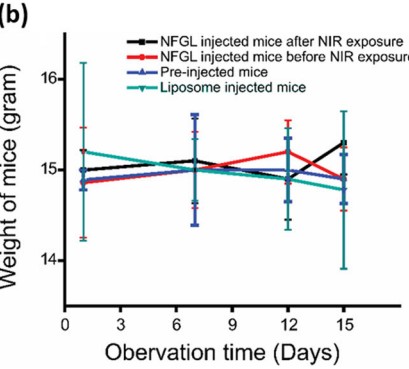

**Fig. 10 In vivo toxicity evaluation of liposomal nanotheranostics. a** Percentage of hemolysis efficacy of liposomes, gold nanoparticles (AuNPs), and graphene quantum dots (GQDs) loaded liposomal nanotheranostics (NFGL) before and after FA attachment at various concentrations (10–200 μg/mL, $n = 3$). **b** Body weight measurements of post-injected various mice groups ($n = 3$). Both the analysis demonstrate the good biocompatibility and safety of engineered liposomal nanotheranostic agents.

heart, (2) normal portal triad, hepatocyte, and central vein in the liver and (3) no acute changes in glomeruli and tubules of the kidney that confirmed the biocompatibility of the injected nanotheranostics agent. Further, hemocompatibility examinations (10–200 μg/mL concentration of nanohybrids) showed negligible hemolysis (below 5% at 100 μg/mL concentration), whereas maximum hemolysis (6–10%) was calculated at the highest concentration (200 μg/mL) of nanohybrids shown in Fig. 10a. Hemolysis procedure was adopted from previously reported methods[6] using DI water and PBS as positive and negative controls. The surface functionalization and functional group masking on the nanohybrid's surface helped in reducing the hemotoxicity. Moreover, the in vivo toxicity/or safety of post-injected nanotheranostic agents was examined by animal health and body weight measurements. The obtained results showed normal health of all animals (pre-injected and post injected) with the controlled body weight demonstrating the better biocompatibility of designed nanohybrids (see Fig. 10b, Supplementary Fig. 18b).

## Discussion

Self-assembled liposomal targeted nanotheranostics platform demonstrated site-specific 4T1 tumor diagnosis and photo-triggered tumor ablation. Enhanced cellular and tumor accumulation of designed nanotheranostics with better contrasting and therapeutic ability were evaluated methodically. Good aqueous dispersion, better colloidal stability (tested up to 1 week), biocompatibility, hemocompatibility, and easy degradation ensured the in vivo investigation of nanotheranostics. Engineered nanohybrids showed imaging bimodality with spatial resolution of solid tumor due to encapsulated dual imaging probes (AuNPs and GQDs) in the liposomal cavity. Post-injected nanohybrids (subcutaneously and intravenously) exhibited strong tumor binding ability (tested up to 48 h) due to the potential targeting effect of FA. Further, surface functionalization revealed the specific biodistribution and smooth circulation of injected nanotheranostics. Enhanced contrast (brightness and fluorescence) of nanohybrids in lower abdomen organs may indicate their easy excretion from the animal body. Moreover, single wavelength NIR light exposure (750 nm, 1 W/cm$^2$) resulted in promising cancer cell death and selective tumor reduction because of generated photothermal heat and ROS (as the side product of PTT[50]). The ROS inhibition was successfully achieved at in vitro and in vivo levels, demonstrating noteworthy cancer cell death and tumor reduction without affecting the surrounding healthy tissue. From in vivo analysis, the surrounding area of tumor

(NIR exposed area) was noticed normally without any eschars and inflammation during NIR light treatment that signposted the ROS scavenging ability of liposomal nanohybrids. Importantly, the combined chemophototherapy effect of nanohybrids exhibited superior tumor reduction compared with stand-alone therapy (chemotherapy and PTT) due to the synergistic effect of generated heat and released anticancer drug. Overall, negligible hemolysis, enhanced cell viability, normal animal health, and controlled body weight confirmed the biocompatibility of designed nanohybrids. Hence, the fabricated liposomal nanotheranostics systems could be a safe and potential platform for multimode tumor diagnostics and therapeutics.

## Methods

**Reagents**. 1,2-Distearoyl-sn-glycero-3-phosphocholine (DSPC) was procured from Lipoid, Switzerland. GQDs were obtained from *Mangifera indica* (known as mango tree in the local campus of IIT Bombay, India). (2′,7′-dichlorofluorescin diacetate, DCFDA), Hydrogen tetrachloroaurate (HAuCl$_4$), 1-ethyl-3-(3-dimethylaminopropyl) carbodiimide (EDC), sodium citrate dehydrate, silver nitrate, cholesterol, trypsin-EDTA, 3-(4,5-dimethylthiazol-2-yl)-2,5-diphenyl-tetrazolium bromide (MTT), N-hydroxysuccinimide (NHS), FA, poly(ethylene glycol) 2-aminoethyl ether acetic acid (COOH–PEG–NH$_2$, M$_n$ 3500), Thiol-PEG-Carboxyl, doxorubicin hydrochloride (DOX.HCL), sodium borohydride (NaBH$_4$, 99%), ascorbic acid (99.5%), and N-cetyltrimethylammonium bromide (CTAB, 99%) were purchased from Sigma-Aldrich Pvt. Ltd., USA. Phosphate-buffered saline (PBS, pH 7.4), fetal bovine serum (FBS), Dulbecco's Modified Eagle Medium (DMEM), dimethyl sulfoxide (DMSO), 4′,6-diamidino-2-phenylindole (DAPI), and antibiotic–antimycotic solution were obtained from HiMedia Pvt. Ltd., India. Millipore (>18.2 MΩ cm) was used for all experiments.

**Characterization techniques**. TEM images were performed with exposure of low-beam voltage (100 kV) using Cryo-mode (FEI Tecnai G2). Fourier transform-infrared spectroscopy (FTIR) was done using 3000 Hyperion Microscope with Vertex 80 FTIR System (Bruker, Germany). Raman spectra were recorded using a Jobin-Yvon Labram spectrometer. Samples were excited with 532 nm laser at 5 mW. Dynamic light scattering and zeta potential measurements were recorded by using (DLS)-BI200SM, Brookhaven Instruments Corporation, USA. Further, UV–vis spectroscopy was performed at a path length of 1 cm using Perkin Elmer Lamda-25. Fluorescence spectroscopy was performed using Shimadzu at a slit width of 5 nm (excitation and emission) in high-sensitivity mode. Atomic force microscopy measurements were recorded by using atomic force microscope (PSIA XE-100) using tapping mode. Fine clean silicon wafers as substrates were used for AFM measurements that were made by drop-casting process. Fluorescence microscopy was carried out by using 465–95, 525–45, and 540–80 nm filters from an inverted fluorescent microscope: Nikon Eclipse TE 2000S. ESR (electron spin resonance) analysis was performed using ESR spectrometer (JES-FA 200). The signals were recorded at room temperature at standard frequency. The ROS measurements were done on a microplate reader. In vivo emissive images were recorded using IVIS spectrum imaging system (IVIS spectrum Xenogen), and contrast images were recorded at clinical CT scanner (TOSHIBA 64 at 120 kVp tube voltage and 250 mA tube current with 5 mm slice thickness and 1 s rotation time). NIR light mediated photothermal transduction experiments were performed

by using 750 nm NIR laser source using 1 W/cm$^2$ power density. All digital pictures were captured by using mobile camera (One Plus 6 T).

**Synthesis of GQDs**. GQDs were synthesized using *Mangifera indica* leaves followed by an earlier reported recipe with some modifications[51]. Small pieces of leaves were dipped in ethanol overnight. The acquired leaf's extract was centrifuged and filtered, that was further concentrated by ethanol evaporation through rotary evaporator. The obtained material was diluted with Milli-Q water and treated under a microwave oven for 5 min (800–900 W). Finally, the prepared residue was dispersed in ethanol and filtered through the syringe filter to obtain GQDs. The prepared final product (GQDs) was dried at 65–70 °C overnight, and stored in dark conditions at room temperature for further usage.

**Synthesis of AuNPs**. Gold nanoparticles were prepared by using a previously reported method with various modifications[47,52]. The AuNPs were prepared by mixing 80 µL of 0.01 M NaBH$_4$, 100 µL of 0.01 M HAuCl$_4$, 140 µL of 0.01 M citrate sodium, and 9.625 mL of ultrapure water under stirring at 2000 rpm. The prepared mixture was added to the solution of 800 µL of 0.01 M HAuCl$_4$, 100 µL of 0.01 M AgNO$_3$, 100 µL of 0.8 M HCl, 400 µL of 0.15 M ascorbic acid, and 50 mL of 0.09 M CTAB solution. The reaction solution was left at 37 °C overnight. After completion of the reaction, the AuNPs were collected via centrifugation (15 K rpm) that were functionalized with the PEG dithiol through surfactant exchange process at room temperature overnight. To obtain the PEGylated AuNPs, 3 mL of CTAB-stabilized AuNPs were incubated with 20 mg of PEG dithiol overnight at room temperature. The surface-modified AuNPs were collected through centrifugation (15 K rpm for 10 min) and washed thoroughly with Milli-Q water, and were dispersed in Milli-Q water for further usage.

**Synthesis of GQD-loaded liposomal nanohybrids**. The preparation method of liposomes was adopted from our earlier reported procedure with some modifications[7]. A mixture of DSPC (80 mg) and cholesterol (20 mg) was dissolved in 15 mL of chloroform at room temperature conditions. After making the complete mixture, the solvent was completely removed at 41 °C. After 1 h of solvent evaporation, 30 ml of PBS containing 1 mg of fluorescent GQDs was added to obtain a phospholipid thin film and kept for further film hydration that was hydrated at 45–47 °C for 60 min using rota evaporator. The hydrated suspension was kept overnight and the next day, having taken to room temperature (37 °C), that was further sonicated (probe sonication of 5 cycles with 40% intensity with 2 s on/off pulse) for 10 min to obtain the small vesicles. The obtained liposomal nanohybrids were characterized through microscopic analysis and used for nanomedicine applications.

**Synthesis of liposomal nanotheranostics (NFGL)**. To design NFGL, a mixture of DSPC (80 mg) and cholesterol (20 mg) was dissolved in 15 mL of chloroform at room temperature, and the solvent was completely removed at 41 °C followed by evaporation process. After obtaining the thin film, 30 mL of PBS containing 1 mg of emissive GQDs and 1 mL of surface-modified AuNPs was added to phospholipid thin film during film hydration process and further hydrated at 45–47 °C for 60 min. The above hydrated suspension was kept overnight and then sonicated for 10 min to obtain the small size vesicles followed by probe sonication (5 cycles with 40% intensity with 2 s on/off pulse). For the synthesis of doxorubicin hydrochloride (DOX.HCL)-loaded NFGL (DOX–NFGL), 0.5 mg/mL DOX was prepared in PBS (pH 7.4). The aqueous solution of DOX was added during thin-film hydration. Further, the remaining procedure was the same as described earlier. The mass of drug loaded in NFGL was calculated by subtracting the mass of drug in the supernatant from the total mass of drug used. Percentage loading efficiency was calculated according to the following equation:

$$\% \text{ Loading} = \frac{\text{Mass of drug in NFGL}}{\text{Mass of NFGL}} \times 100 \tag{1}$$

**Surface functionalization of NFGL with FA**. FA (150 mg) was activated by 1-ethyl-3-(3-dimethylaminopropyl) carbodiimide (EDC, 60 mg) and N-hydroxysuccinimide (NHS, 40 mg) in 45 mL of Milli-Q water for 24 h. The reaction mixture was protected from light. In all, 5 mL of 0.5 mg/mL aqueous solution of amine functionalized polyethylene glycol (COOH–PEG–NH$_2$) was mixed with activated FA and allowed to react for 12 h. The PEGylated FA (1 mg/mL) as targeting ligand was attached on the surface of NFGL (5 mg/mL) through incubation process at room temperature. After completion of the reaction, the products were dialyzed for 24 h.

**Aqueous dispersibility, degradation, and photostability**. Synthesized nanohybrids (1 mg/mL) were mixed in aqueous suspension that was observed for 24 h of incubation time at ambient conditions and laser exposure (750 nm, 1 W/cm$^2$ for 10 min). During dispersibility test, the digital photographs were captured at 1 h and 24 h of time periods. Further, the photostability of NFGL (1 mg/mL) was checked up to 2 days under 30 min of intermittent exposure of NIR (750 nm) and UV light (365 nm). The prepared NFGL–FA nanotheranostics was treated in various

conditions, such as NIR light exposure, mixing in acidic condition, and exposed with NIR light (combination therapeutic condition) to evaluate the degradation. In addition, the aqueous suspension of NFGL, DOX–NFGL, and GQDs–liposomes was prepared. These prepared suspensions were kept at room temperature for up to 1 week time period to observe the turbidity and aggregation. Digital photographs of treated NFGL were captured at various time points (1st, 2nd, 3rd, and 7th day).

**Photothermal transduction assessment**. Various concentrations (0.2–1.0 mg/mL) of NFGL in PBS were prepared for photothermal transduction experiments. The surrounding temperature (water bath) was stabilized to 37 °C. About 100 µL of NFGL and GQDs-loaded liposomes were added into 96-well plates and exposed to 750-nm (1 W/cm$^2$ power) continuous wave (CW) NIR laser source for 10 min. Time-dependent photothermal response was recorded by a digital thermometer. PBS and parent liposomes were used as controls.

**ROS analysis**. ROS production from NFGL was analyzed using DCFDA (2′,7′-dichlorofluorescin diacetate) dye staining. The designed nanohybrids were treated with NIR light, and ESR (electron spin resonance) analysis was performed using ESR spectrometer (JES-FA 200). About 750 nm light was irradiated over NFGL dispersed in water, and signals were recorded at standard frequency (8.75–9.65 GHz) at room temperature. To examine the ROS from cancer cell lines in the therapeutic conditions, 4T1 and MCF-7 cells were seeded at the density of 2 × 10$^4$ in 96-well plates, and 100 µL of 1 mg/mL NFGL was added and left for 24 h of incubation. The next day, these cells were irradiated with 750 nm light for 10 min. After treatment, the fluorescence (excitation: 495 nm, emission: 520 nm) was measured at different time intervals after adding DCFDA (10 µM) dye. In these experiments, various phases of NFGL, such as NFGL treated cells before NIR exposure, GQDs loaded liposomes with NIR irradiation, and NFGL treated cancer cells after NIR light exposure, were tested to ensure the ROS generation during PTT measurements, and scavenging the ROS by using engineered NFGL nanohybrids in a real situation of phototherapy.

**Stimuli-triggered drug release**. To check the kinetics of drug release in response to NIR light, 2 mL of DOX–NFGL was added in a dialysis bag immersed in 200 mL of PBS at various pH (3 and 7.4). At various time intervals, and before and after NIR light irradiation, 2 mL of solution was collected and replaced with the same volume of fresh PBS solution. The amount of DOX released was measured with the help of the following equation:

$$\% \text{ Release} = \frac{\text{Mass of drug at time}(t)}{\text{Initial mass of drug}} \times 100 \tag{2}$$

**In vitro biocompatibility**. To check the biocompatibility of NFGL and its variants, NIH-3T3 cells were seeded at a density of 2 × 10$^4$ cells per well in 96-well plates and incubated for 24 h in 5% CO$_2$ atmosphere at 37 °C using Dulbecco's Modified Eagle's Medium (DMEM, Gibco, Carlsbad, CA, USA) supplemented with 10% FBS and penicillin/streptomycin. In MTT assay, after 24 h of incubation, 100 µL of different concentrations (0.1–1 mg/mL) of NFGL, FA attached NFGL, GQDs, and surface modified AuNPs were added into the wells. Following 24 h of incubation, wells were washed off with PBS, and 20 µL of MTT dye was added. Formazan crystals formed after 4 h were dissolved using 200 µL of DMSO. Optical absorbance was recorded at 570 and 690 nm using microplate reader (Tecan Infinite 200 PRO). The percentage of cell viability was calculated in reference to untreated cells (control).

**In vitro cellular uptake and ROS measurements**. Breast cancer cells were cultured in DMEM culture media that was supplemented with 10% FBS and penicillin/streptomycin. To check the targeting ability of NFGL, 4T1 cells were seeded into 12-well plates at a density of 2 × 10$^3$ cells/well and incubated for 24 h in 5% CO$_2$ atmosphere at 37 °C. After being rinsed with PBS, 100 µg/mL of NFGL–FA nanoparticles were added and treated in the conditions of with and without NIR light (750 nm NIR light exposure was for 10 min). After 6 h of incubation, cells were washed with PBS three times to get rid of all the unbound particles. About 4% paraformaldehyde solution was added to the cells, left for 10 min, and then nuclei were stained with 4,6-diamidino-2-phenylindole (DAPI, 1 µg/mL in PBS). At the end of the incubation period, the staining solution was repeatedly washed with PBS. The coverslip was mounted on a drop of 70% glycerol on a glass slide, and fluorescence images were taken using fluorescence microscope. To evaluate the in vitro ROS from cancer cellular environment, 4T1 cells were treated with NFGL and GQDs loaded liposomal nanohybrids in the presence and absence of NIR light irradiation. After 10 min of NIR light exposure and additional 6 h of incubation, these cells were stained with DCFDA (2′,7′-dichlorofluorescin diacetate) dye. The green emission intensity from the cells were recorded after adding DCFDA (10 µM).

**In vitro therapeutic and ROS scavenging performance**. 4T1 breast cancer cells were seeded into 96-well plates at a density of 2 × 10$^4$ cells/well and left overnight in 5% CO$_2$ atmosphere at 37 °C. After rinsing the wells with PBS, cells were

incubated with several components of NFGL, such as FA functionalized GQDs (GQD-FA) and NFGL (NFGL-FA), and DOX loaded NFGL–FA nanohybrids with and without NIR light (750 nm, 1 W/cm$^2$) treatment. After 6 h of incubation, wells were rinsed with PBS three times to get rid of all the unbound particles. After treatment, these plates were incubated for another 24 h. Formazan crystals formed after 4 h were dissolved using 200 μL of DMSO. Optical absorbance was recorded at 570 nm and 690 nm using microplate reader (Tecan Infinite 200 PRO). The percentage of cell viability was calculated in reference to untreated cells (control). Similarly, MCF-7 cells were seeded into 96-well plates at a density of $2 \times 10^4$ cells/well and left overnight in 5% $CO_2$ atmosphere at 37 °C. After rinsing the wells with PBS, cells were incubated with several variants of NFGL and controls (GQDs). After 6 h of incubation, the wells were rinsed with PBS three times to get rid of all the unbound particles. The following groups were categorized for comparative therapeutic assessment—group 1: only cells, group 2: NIR light exposed cells, group 3: only GQDs treated cells, group 4: GQDs-FA treated cells, group 5: GQDs–FA treated cells and exposed to 10 min of NIR light, group 6: NFGL treated cells, group 7: NFGL–FA treated cells, group 8: NFGL–FA treated cells and exposed to 10 min of NIR light (targeted phototherapy), group 9: DOX–NFGL treated cells, group 10: DOX–NFGL–FA treated cells (targeted chemotherapy), and group 11: cells treated with DOX–NFGL–FA along with 10 min of NIR light exposure (targeted combined chemo-PTT). After treatment, the plates were incubated for another 24 h; thereafter, MTT assay was performed as described above.

**In vivo tumor growth measurements**. Experimental protocols on Balb/c mice were approved by the Institutional Animal Ethical Committee (IAEC) of the National Centre for Cell Science, Pune, India (NCCS, Pune). The IAEC allows us to conduct the in vivo experiments as per institute guidelines according to the National Centre for Cell Science, Pune, India's laws. IAEC's laws approved that all experiments were performed in completion with the guidelines of the IAEC research program under B318 project. In vivo examinations were followed by using our earlier reported methodology[7]. Six-week old female Balb/c mice were used for the present study. In total, $2 \times 10^5$ 4T1 breast cancer cells were injected subcutaneously into the mammary fat pad of Balb/c mice, and tumor growth evaluated.

**Image-guided tumor uptake and biodistribution studies**. Due to multimode-contrasting probes (emissive GQDs and contrasting AuNPs) in NFGL, the designed system was able to demonstrate its imaging bimodality with breast tumor model. A one-time dose (10 mg/kg body weight) of targeted NFGL–FA nanohybrids was subcutaneously injected into 4T1 tumor site of female Balb/c mice. After 1 h of post injection, mice were exposed for 10 min of NIR light (750 nm, 1 W/cm$^2$); then the in vivo NIRF images of treated animals were captured at various time points (1, 6, and 48 h) in the anesthesized condition using IVIS. These mice were compared with non-NIR light exposed and pre-injected mice group (control). The emissive intensity (xenogen for NIRF at 500 nm of excitation) from whole body luminescence and major organs was measured to examine the in vivo biodistribution of post-injected animals at various time points (1, 6, and 48 h). Similarly, time-dependent X-ray computed tomography (X-ray CT) images were captured using clinical scanner (TOSHIBA 64 CT scanner) at 120 kVp tube voltage and 250 mA tube current with 5 mm slice thickness and 1 s rotation time. A single dose (10 mg/kg) of NFGL–FA was subcutaneously injected into 4T1 tumor site of female Balb/c mice. Post-injected mice were exposed for 10 min of NIR light (750 nm, 1 W/cm$^2$); then the in vivo CT scans of treated animals were captured at various time points (1, 6, and 48 h) in the anesthesized condition. These scans were compared with non-NIR light exposed and pre-injected mice group (control). The brightness from the whole body and major organs was measured to examine the in vivo biodistribution of post-injected animals at various time points (1, 6, and 48 h). On the other hand, a minimal dose (10 mg/kg body weight) of NFGL–FA nanohybrids was intravenously injected into tumor bearing female Balb/c mice. In vivo NIRF images of NFGL–FA nanohybrid injected animals were captured at various time points (1, 24, and 48 h) using IVIS. The biodistribution of injected fluorescent nanohybrids was measured with major organs of tumor bearing mice at different time points of post injection, as mentioned above. Further, after 48 h of post administration, treated animals were sacrificed, and all major organs were collected and used for ex vivo NIRF imaging.

**Near-infrared light mediated tumor reduction**. To observe the therapeutic outcomes (antitumor activity), different formulations of designed nanohybrids, such as NFGL–FA, DOX–NFGL–FA, and GQDs–liposome–FA, were tested on 4T1 tumor-bearing mice with and without NIR light exposure. A total of five sets of animal groups were prepared (3 mice per group) as follows: (1) control animal group (only pre-injected and untreated animals), (2) DOX–NFGL–FA injected animals (tumor reduction by targeted chemotherapeutic effect that was without NIR light treatment), (3) DOX–NFGL–FA injected animals (tumor reduction by targeted chemo–photothermal therapeutic effect that was under NIR light treatment), (4) GQD–liposome–FA injected animals (tumor reduction by produced ROS and generated heat under NIR light irradiation), and (5) NFGL–FA injected animals (tumor reduction by generated heat under NIR light irradiation where produced ROS was scavenged by the nanoparticles). Once the tumor size was

notable, the above formulations of fabricated nanohybrids with minimal dose (10 mg/kg body weight, 100 μL) were intravenously injected into 4T1 tumor bearing female Balb/c mice. The nanohybrid injected 4T1 tumor-bearing animals were treated with 750 nm of NIR light (1 W/cm$^2$ power of laser irradiation for 10 min). A total of four therapeutic courses were conducted with a time interval of 2 days followed by the same conditions mentioned above. Animal health, tumor volume, and body weight were measured during therapeutic courses. After completion of the therapeutic course, all mice were sacrificed, and all major organs and tumors were collected for ex vivo and histopathological examinations. Digital photographs were captured during these treatment days.

**In vivo toxicity measurements**. Histopathological examinations (hematoxylin and eosin, H&E at 6 and 48 h), body weight measurements, and health behavior of treated animals were examined to ensure the adverse effect of injected liposomal based NFGL–FA nanotheranostics. Further, hemolysis measurements of engineered nanohybrids were carried out with blood samples of treated animals that confirm the in vivo toxicity. On the 36th day with respect to the initial cell culture day (considered as day 0), post-administered animals were sacrificed, and all major organs were collected for hematoxylin and eosin validations, and to investigate the tissue injury. During whole treatment planning, the body weight of all treated animals was observed and compared with the control animal group. In hemolysis study, we followed the previously reported method with some modifications[6]. Healthy animals were used to collect 0.5 mL of blood in ethylenedia-minetetraacetic acid-stabilized tubes that were further centrifuged and washed with PBS. Further, 0.1 mL of collected red blood cells were diluted with 4 mL of PBS. In all, 500 μL volumes of RBCs were treated with various concentrations (10–200 μg/mL) of nanohybrids, such as liposomes, NFGL, and FA attached NFGL. Treated nanohybrids were incubated with RBCs for a period of 6 h at room temperature (RT), and then the mixtures were centrifuged at 5000 rpm for 10 min. UV–Vis absorption spectroscopy was used to calculate the hemolysis of treated nanoparticles through observing the released hemoglobin into the solution from hemolyzed RBCs (absorbance of hemoglobin at 540 nm). The percentage of hemolysis was calculated using the equation: % hemolysis = [(absorbance of the used sample − absorbance of negative control)/(positive control absorbance − negative control absorbance)] × 100.

**Statistics and reproducibility**. Statistical measurements of all experiments were demonstrated in triplicate. Graphs were plotted by using OriginPro 8 and sigma plot 10.0 software. For in vivo studies, experimental protocols with samples of animals ($n = 3$ mice per group and total $n = 5$ treatment groups) on 6-week-old Balb/c female mice were approved by the Institutional Animal Ethical Committee (IAEC) of the National Centre for Cell Science, Pune, India (NCCS, Pune). The IAEC allows us to conduct the animal experiments as per institute guidelines according to the National Centre for Cell Science, Pune, India's laws. IAEC's laws approved that all experiments were performed in completion with the guidelines of the IAEC research program under B318 project. All cell lines were from American Type Culture Collection (ATCC) that were authenticated and cultured at the National Centre for Cell Science, Pune. Remarkable observations between different groups were assessed by $t$ test.

**Reporting summary**. Further information on research design is available in the Nature Research Reporting Summary linked to this article.

## Data availability
Supporting data for the present study are available within this article or in the Supplementary Information file, and other data information are available from the authors on their reasonable request.

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

## Acknowledgements

The authors acknowledge Sophisticated Analytical Instrumentation Facility (SAIF) at IIT Bombay for instrumentation support. N.K.J. and J.D. would like to thank DST-INSPIRE fellowship. This work was supported by the Department of Biotechnology, Government of India. We are grateful to the National Center for Cell Science, Pune, for in vivo studies. J.C. acknowledges ERC-2019-STG. We are thankful to Mr. Anand and Dr. Ashish Kumar (MD, medical radiologist) for fruitful discussion on tomography imaging and analysis.

## Author contributions

R.P., N.K.J., and R.S. have designed the project and experiments. M.K.K. has synthesized the GQDs, R.P. and N.K.J. have engineered the liposomal nanohybrids, J.D. and S.S. have performed the encapsulation experiments and characterizations of nanohybrids, D.S.C. and M.G. have conducted the in vitro studies, and M.G., A.S.Y., and G.C.K. have demonstrated the in vivo studies. R.P., A.S.T., J.C., and N.K.J. wrote the paper. N.K.J and J.C. have analyzed the in vitro outcomes, and R.P. and J.C. have designed all the schematics for this work. Finally, all authors have contributed for the preparation of the final version of the paper and given approval to the final paper.

## Competing interests

The authors declare no competing interests.
