## [Peer Review File · Communications Biology]

Reviewers' comments:

Reviewer #1 (Remarks to the Author):

In this work, the authors develop a graphene-gold-liposome material and use it for photothermal therapy. This combination of 3 well-explored materials is of interest, and the imaging results are interesting. However, the authors should go a bit further in demonstrating the potential of the material:

- 1) An anti-tumor study should be carried out
- 2) Cryo-EM images of the material should be provided.
- 3) In the schematic, graphene is pointing outward into the aqueous environment, but is presumably hydrophobic.

Reviewer #2 (Remarks to the Author):

In this work, a multifunctional liposome based nanohybrids loaded with gold nanoparticles (GNPs) and emissive graphene quantum dots (GQDs) were engineered. The nanohybrids exhibited dual-modal imaging for in vivo tumor diagnosis along with specific biodistribution and long-time tumor binding ability. Enhanced cell uptake and ROS scavenging ability of the nanohybrids were observed during single wavelength near infra-red (NIR) light (750 nm) irradiation. However, there are still many major concerns in this paper listed as following:

1. In this study, the synthesized nanohybrids were injected subcutaneously into the tumor site. Although the imaging results were impressive, its clinical application value was difficult to evaluate. In addition, although folic acid is coupled to target tumor, the safety and distribution of the nanohybrids are still evaluated by subcutaneous injection, and its biological effect is difficult to be evaluated by existing data. It would be more convincing to evaluate the performance of the nanohybrids by the way of intravenous injection.
2. The authors reported that the nanohybrids combined with doxorubicin hydrochloride killed tumor cells and inflammation in vitro, but no experiment was verified in vivo. It is suggested to perform the experimental data of in vivo treatment of the nanohybrids to make the whole study be complete and more logical.
3. Line385 The contrast and emission enhancement in lower abdomen organs may reveal the clearance of small sized imaging agents (GQDs and AuNPs), however, it can also cause organ toxicity to lower abdomen organs such as the spleen and intestine. Therefore, more stringent biotoxicity tests other than histopathology analysis should be performed to determine whether the nanohybrid has good biocompatibility.
4. Enhanced scavenging behavior of nanohybrid during NIR light exposure should also be demonstrated in vivo.

Minor concerns:

1. Line123 The full name of the term "TEM" first appearing in the text should be provided. Line 124 of the paper mentions that the size of the engineered liposomal nanohybrid is in the range of 200-250 nm. However, Figure 2 shows that the size and morphology of the liposomes are obviously not uniform, which do not match the description of "Uniformly distributed parent liposomes were shown in Figure 2d with controlled and spherical morphology."
2. The results of EDAX shown in Fig. S3 are not clear enough to distinguish the elements represented by each peak. At first glance, it seems difficult to distinguish whether some peaks are Au or Ag, although there is no Ag in theory.
3. Figure2 and Figure S9 The font size in the lower left corner of the picture should be the same.
4. In the element mapping analysis shown in Figure 3, the single element map on the left and the

merged map on the right are not well integrated. What elements do the blue dot, green dot and red dot in the right figure correspond to respectively?

5. Line 218 The expression of "room temperature (37°C)" is inappropriate, as the concept of room temperature usually refers to 25°C.

4. What do (a and (b in Figure 4b inset represent?

5. What does "Cells+L" mean in Figure S15? Also, p-value should be given in Figure S14 and S15.

6. The quality of the images in this paper is suggested to be improved. For example, in Figure 6 c, d, the resolution of the text in the figure needs to be increased. And the superscript of Figure 6d is wrong. In Figure 7, the resolution of the text in the figure needs to be increased and the font size should be adjusted.

Response to reviewer's comments on the manuscript entitled, "*Stepwise Assembly of Multimode Liposomal Nanotheranostic Agent for Targeted In Vivo Bioimaging and Near-Infrared Light Mediated Cancer Therapy*" (Manuscript ID: COMMSBIO-19-1711-T)

We thank the reviewers for their suggestions and comments on the manuscript. All the comments of all reviewers have been addressed in the major revision of the manuscript as suggested. The revised manuscript includes additional data and their results and discussion generated from freshly conducted experiments as per reviewer's comments. Further, changes in data presentation and addition of explanations have been incorporated at appropriate places in the manuscript. The anti-tumor activity of designed nanohybrids was highly recommended by the reviewers that is addressed in the revised manuscript. Overall, the single nanohybrid is capable for site selective tumor diagnosis and localized tumor reduction (named as nanotheranostic agent), and these abilities have been highlighted in the present manuscript. Due to explanation of antitumor performances of designed system, we have modified the manuscript title from "Stepwise Assembly of Multimode Liposomal Nanocontrast Agent for Targeted *In Vivo* Bioimaging" to "*Stepwise Assembly of Multimode Liposomal Nanotheranostic Agent for Targeted In Vivo Bioimaging and Near-Infrared Light Mediated Cancer Therapy*". Now, we believe that the present manuscript is better understandable for the readers. We request the reviewers to kindly go through the manuscript.

Our responses to the reviewer's comments along with the details of various changes made in the manuscript are as following.

Reviewer 1

Comments to the Author

"In this work, the authors develop a graphene-gold-liposome material and use it for photothermal therapy. This combination of 3 well-explored materials is of interest, and the imaging results are interesting. However, the authors should go a bit further in demonstrating the potential of the material:"

Our response: Our sincere thanks to reviewer for the appreciation and finding the importance of present manuscript. We have evaluated the potential impact of designed materials for localized imaging and cancer therapy which are addressed in the revised manuscript with highlights. We request reviewer to go through the revised manuscript.

Comment 1: “An anti-tumor study should be carried out.”

Our response: Our thanks to the reviewer for suggesting the anti-tumor study of engineered nanohybrids. Anti-tumor study of our materials has been demonstrated on 4T1 tumor bearing Balb/c mice models. In brief: five groups of animals (3 mice per group) have prepared and tested with various therapeutic conditions to evaluate the anti-tumor activity of designed nanotheranostic agents. Intravenous administration (10 mg/kg body weight) of different formulations of engineered nanohybrids such as folic acid attached NFGL (NFGL-FA), doxorubicin hydrochloride loaded NFGL-FA (DOX-NFGL-FA) and GQDs loaded liposomes with folic acid attachment have been used for phototriggered anti-tumor therapy and tumor ablation. During *in vivo* therapeutic course, various qualitative and quantitative analysis of tumor volume and tumor weight are evaluated. A drastic reduction in tumor volume and weight is measured in the case of DOX-NFGL-FA (39.04 mm³ and 0.041 gram) during NIR light exposure treatment due to the combined effect of phototriggered release of an anticancer drug and generated heat in tumor environment that is quite comparable with GQDs-Liposome-FA (82.07 mm³ and 0.085 gram) treated mice due to the combined effect of produced ROS and heat (key factor of photothermal therapy) in tumor environment as shown in Figure 9d and Figure 9e. Interestingly, the GQDs-Liposome-FA nanohybrid induce the oxidative and thermal damage for significant tumor ablation/reduction. Additionally, designed NFGL-FA nanohybrids showed the significant tumor shrinking in terms of tumor volume and tumor weight (120.35 mm³ and 0.070 gram) as compared to control animal group (142.07 mm³ and 0.119 gram pre-treated animals) due to the promising effect of generated photothermal heat as validated with the obtained data given in the Figure 9d and Figure 9e. Overall, in the present manuscript, we have incorporated the obtained data and their analyzed results and discussion for *in vivo* therapeutic outcomes of designed nanohybrids. The obtained data have been presented as a part of Figure 9 which is given in the revised manuscript. We request reviewer to please go through the highlighted part and newly obtained data incorporated in the revised manuscript.

Comment 2: “Cryo-EM images of the material should be provided.”

Our response: We thank to the reviewer for this constructive suggestion. As per reviewer's comment, we have prepared fresh samples of nanohybrids and characterized through Cryo-mode (FEI Tecnai G2) Transmission Electron Microscopic imaging at low beam voltage (100 kV). Obtained Cryo-EM images of the material are shown as part of Figure 2 in the main article of revised manuscript and Figure S1 in the supporting information. We request reviewer to please go through the newly obtained images incorporated in the revised manuscript.

Comment 3: “In the schematic, graphene is pointing outward into the aqueous environment, but is presumably hydrophobic.”

Our response: Our thanks to the reviewer for the query. Prepared graphene quantum dots are hydrophilic in nature that are loaded into the cavity of liposomes. We have modified the schematic and have followed reviewer’s suggestions which are addressed at appropriate places in the manuscript.

Reviewer: 2

Comments to the Author

In this work, a multifunctional liposome based nanohybrids loaded with gold nanoparticles (GNPs) and emissive graphene quantum dots (GQDs) were engineered. The nanohybrids exhibited dual-modal imaging for *in vivo* tumor diagnosis along with specific biodistribution and long-time tumor binding ability. Enhanced cell uptake and ROS scavenging ability of the nanohybrids were observed during single wavelength near infra-red (NIR) light (750 nm) irradiation. However, there are still many major concerns in this paper listed as following:

Our response: We would like thank the reviewer for commenting on the impact of designed nanohybrid. Yes, we have demonstrated the enhanced cell uptake and ROS scavenging ability of the designed nanohybrids during single wavelength near infra-red (NIR) light (750 nm) irradiation. We have addressed reviewer’s queries and major concerns in the present manuscript as explained below

Comment 1: “In this study, the synthesized nanohybrids were injected subcutaneously into the tumor site. Although the imaging results were impressive, its clinical application value was difficult to evaluate. In addition, although folic acid is coupled to target tumor, the safety and distribution of the nanohybrids are still evaluated by subcutaneous injection, and its biological effect is difficult to be evaluated by existing data. It would be more convincing to evaluate the performance of the nanohybrids by the way of intravenous injection.”

Our response: Our thanks to the reviewer for the suggestion. We have conducted a comprehensive *in vivo* study with various fresh experiments for targeted tumor imaging and specific bio-distribution in 4T1 tumor bearing Balb/c mice using intravenous injection of folic acid functionalized NFGL (Graphene Quantum Dots and Gold nanoparticles encapsulated liposomes) nanohybrids. Intravenously administrated NFGL-FA demonstrate the site selective tumor diagnosis at various time points (1 h, 24 h and 48 h) confirmed by near infrared *in vivo* imaging of animals. After 48 h of post-injection, specific bio-distribution of injected nanohybrid has been analyzed with significant emissive intensity from major organs and tumor. Further, NIR light mediated tumor reduction study has been demonstrated on 4T1 tumor bearing Balb/c

mice followed by Intravenous administration of designed nanotheranostics (NFGL-FA and its various formulations). To examine the antitumor activity of nanohybrids, five groups of animals (3 mice per group) have prepared and tested with various therapeutic conditions. Intravenous administration (10 mg/kg body weight) of different formulations of engineered nanohybrids such as folic acid attached NFGL (NFGL-FA), doxorubicin hydrochloride loaded NFGL-FA (DOX-NFGL-FA) and GQDs loaded liposomes with folic acid attachment have been used for phototriggered anti-tumor therapy and tumor ablation with and without NIR light (750 nm, 1 W power) exposure. Significant observations of these treatment strategies have been incorporated and highlighted in the present manuscript. Obtained data of *in vivo* tumor diagnosis and therapeutics have been presented as part of Figure 9 in the main article and Figure S17 in the supporting information. We request reviewer to please go through the highlighted part and newly obtained data incorporated in the revised manuscript.

Comment 2: “The authors reported that the nanohybrids combined with doxorubicin hydrochloride killed tumor cells and inflammation *in vitro*, but no experiment was verified *in vivo*. It is suggested to perform the experimental data of *in vivo* treatment of the nanohybrids to make the whole study be completer and more logical.”

Our response: We thank to the reviewer for this suggestion. We have conducted fresh experiments for targeted tumor reduction using intravenous injection of doxorubicin hydrochloride loaded folic acid functionalized NFGL (Graphene Quantum Dots and Gold nanoparticles encapsulated liposomes) nanohybrids. Doxorubicin hydrochloride loaded folic acid attached NFGL (DOX-NFGL-FA) showed significant tumor reduction as compared to control animal group (pre-injected animals) indicating the potential impact of targeted chemotherapy. Newly obtained data of *in vivo* tumor therapeutics have been presented as part of Figure 9 in the main article. Analyzed results and discussion of targeted chemotherapeutic effect on 4T1 tumor regression have been incorporated at appropriate places in the manuscript.

Comment 3: “Line385 The contrast and emission enhancement in lower abdomen organs may reveal the clearance of small sized imaging agents (GQDs and AuNPs), however, it can also cause organ toxicity to lower abdomen organs such as the spleen and intestine. Therefore, more stringent biotoxicity tests other than histopathology analysis should be performed to determine whether the nanohybrid has good biocompatibility.”

Our response: We thank to the reviewer for this query. Yes, it fact that we have observed the contrast and emission intensity in lower abdomen organs that may reveal the clearance of small sized injected imaging agents. To evaluate the *in vivo* toxicity of designed nanohybrids, we have demonstrated hemolysis, mice health and body weight observations that are apart from histopathology analysis. A negligible hemolysis (below 5 %) has been noticed at higher concentration of nanohybrid (100 µg/mL). Additionally, various formulations of designed

nanohybrid have been administrated into tumor bearing mice body showing significant tumor reduction with affecting the surrounding healthy tissues. Interestingly, we have not observed any symptoms of eschars and inflammation on animal's body during treatment strategies (with and without NIR light irradiation) demonstrated the good biocompatibility of injected nanotheranostic agents. Moreover, controlled body weight and good animal health of all treated animals corroborated the better biocompatibility of designed nanohybrids. Obtained data have been presented as part of Figure 10 in the main article and Figure S18 in the supporting information. Results and discussion of *in vivo* toxicity (nanohybrids treated animals) have been incorporated at appropriate places in the manuscript. We request reviewer to please go through the given details in the manuscript.

Comment 4: "Enhanced scavenging behavior of nanohybrid during NIR light exposure should also be demonstrated *in vivo*."

Our response: We thank the reviewer for the suggestion. Fresh *in vivo* experiments (3 mice per group) have been accompanied for enhanced ROS scavenging ability of folic acid functionalized NFGL (Graphene Quantum Dots and Gold nanoparticles encapsulated liposomes, a scavenger system) and GQDs encapsulated Liposomal (GQDs-Liposome-FA) nanohybrids. Intravenously administrated GQDs-Liposome-FA nanohybrid demonstrate the significant tumor reduction (82.07 mm³ volume and 0.085 gram weight) during NIR light exposure (750 nm, 1 W power) due to the combined effect of produced ROS and heat in tumor environment, whereas folic acid functionalized NFGL nanohybrids showed slightly low tumor shrinking (120.35 mm³ and 0.070 gram) due to the standalone effect of only generated photothermal heat that confirmed the scavenging ROS produced during NIR light exposure. More importantly, there were no symptoms of eschars and inflammation on animal's body that are generally observed during NIR light treatment indicated the ROS scavenging ability of treated NFGL-FA nanohybrid. We have incorporated the reviewer's suggestion and obtained data are added as part of Figure 9 in the main article. Analyzed results and discussion are highlighted in the manuscript.

Minor concerns:

Comment 1: "Line123 The full name of the term "TEM" first appearing in the text should be provided".

"Line 124 of the paper mentions that the size of the engineered liposomal nanohybrid is in the range of 200-250 nm. However, Figure 2 shows that the size and morphology of the liposomes are obviously not uniform, which do not match the description of "Uniformly distributed parent liposomes were shown in Figure 2d with controlled and spherical morphology."

Our response: We thank to the reviewer for these suggestions. We have provided the full name of the term "TEM" in the manuscript. Additionally, we have prepared fresh samples of

engineered nanohybrids and characterized freshly with various techniques. Obtained data have been incorporated in the manuscript. Now, the analyzed results and their discussion have been modified with appropriate explanations. Overall, we have incorporated reviewer's suggestions which are highlighted in the manuscript.

Comment 2: "The results of EDAX shown in Fig. S3 are not clear enough to distinguish the elements represented by each peak. At first glance, it seems difficult to distinguish whether some peaks are Au or Ag, although there is no Ag in theory".

Our response: We are grateful to the reviewer for this suggestion. We have revised the EDAX spectrum and improved the Figure S3. Now, we believe that peaks are clear enough to distinguish the elements representation. On the other hand, we have used the surfactant directed process for gold nanoparticles preparation which is addressed in the present manuscript. The synthesis recipe of gold nanoparticles describe the mixing of silver nitrate (AgNO₃) with HAuCl₄ (a gold precursor) in the presence of ascorbic acid as a reducing agent which satisfy the theory of Ag in EDAX spectrum. We have followed reviewer's suggestion and have incorporated the theory of Ag in the revised manuscript.

Comment 3: "Figure2 and Figure S9 The font size in the lower left corner of the picture should be the same".

Our response: We are grateful to the reviewer for this suggestion. We have revised the Figure 2 and Figure S9 followed by reviewer's suggestion and have maintained the same font size in the lower left corner of the pictures.

Comment 4: "In the element mapping analysis shown in Figure 3, the single element map on the left and the merged map on the right are not well integrated. What elements do the blue dot, green dot and red dot in the right figure correspond to respectively?"

Our response: We thank to the reviewer for this suggestion. We have revised the Figure 3 and have provided the explanation of each element with color dot individually given in the right figure.

Comment 5: "Line218 The expression of "room temperature (37°C)" is inappropriate, as the concept of room temperature usually refers to 25°C."

Our response: We thank to the reviewer for this suggestion and explanation. We have revised the details of room temperature concept given in the manuscript and have incorporated reviewer's suggestion.

Comment 6: “What do (a and (b in Figure 4b inset represent?”

Our response: We would like to thank the reviewer for this query. Absorption and photoluminescence spectroscopic measurements of designed nanohybrids and its components are given in the Figure 4a and Figure 4b, respectively. We have revised the Figure 4b and replaced the picture (a digital photograph of sample under day light and UV exposure) given in inset. Now, we believe that Figure 4a and Figure 4b are clear and understandable.

Comment 7: “What does "Cells+L" mean in Figure S15? Also, p-value should be given in Figure S14 and S15.”

Our response: We would like to thank the reviewer for the query. NIR light treated cells has been mentioned as Cells+L in Figure S15 that is now modified to Cells+NIR. We have revised Figure S14 and Figure S15, and have given the p-values in the Figures.

Comment 8: “The quality of the images in this paper is suggested to be improved. For example, in Figure 6 c, d, the resolution of the text in the figure needs to be increased. And the superscript of Figure 6d is wrong. In Figure 7, the resolution of the text in the figure needs to be increased and the font size should be adjusted.”

Our response: We thank to the reviewer for this suggestion. We have improved the quality of all images which are incorporated at the appropriate place in the manuscript. Further, the resolution of the text in the Figure 6 c, d has been improved and the superscript of Figure 6d is corrected. Similarly, in Figure 7, the resolution of the text and the font size have been adjusted as advised by reviewer.

REVIEWERS' COMMENTS:

Reviewer #1 (Remarks to the Author):

The authors have made a nice effort to revise the manuscript and I appreciate their efforts. I recommend publication of this work.

Reviewer #2 (Remarks to the Author):

The authors have addressed all my concerns. I think it can be accepted by Communications Biology.